# The Graph Cut Kernel for Ranked Data

**Michelangelo Conserva** *m.conserva@qmul.ac.uk*
*Department of EECS*
*Queen Mary University of London*

**Marc Peter Deisenroth** *m.deisenroth@ucl.ac.uk*
*UCL Centre for Artificial Intelligence*
*University College London*

**K S Sesh Kumar** *s.karri@imperial.ac.uk*
*Data Science Institute*
*Imperial College London*

**Reviewed on OpenReview:** *https://openreview.net/forum?id=SEUGkraMPi*

## Abstract

Many algorithms for ranked data become computationally intractable as the number of objects grows due to the complex geometric structure induced by rankings. An additional challenge is posed by partial rankings, i.e. rankings in which the preference is only known for a subset of all objects. For these reasons, state-of-the-art methods cannot scale to real-world applications, such as recommender systems. We address this challenge by exploiting the geometric structure of ranked data and additional available information about the objects to derive a kernel for ranking based on the graph cut function. The graph cut kernel combines the efficiency of submodular optimization with the theoretical properties of kernel-based methods. We demonstrate that our novel kernel drastically reduces the computational cost while maintaining the same accuracy as state-of-the-art methods.

## 1 Introduction

Ranked data arises in many real-world applications, e.g., multi-objects tracking (Hu et al., 2020) and preference learning (Fürnkranz & Hüllermeier, 2003). However, the classical application is recommender systems (Karatzoglou et al., 2013), for example movies (Harper & Konstan, 2015) or jokes (Goldberg et al., 2001). In such scenarios, rankings can take different forms (full, top-$k$ or interleaving) and, typically, contain information only for a subset of the objects. The main challenge of machine learning applications for ranked data is computational as rankings present a super exponential growth. Given $n$ objects, there are $n!$ possible full rankings and about $\frac{1}{2}(\frac{1}{\log(2)})^{n+1}n!$ partial rankings (Gross, 1962).

Several approaches have been proposed to model ranked data. Balcan et al. (2008) and Ailon et al. (2008) propose to interpret rankings as a sequence of binary decisions between objects whereas Lebanon & Mao (2008), Helmbold & Warmuth (2009) and Huang et al. (2009) define a distribution over permutations. These approaches, however, do not scale when the number of objects is large since they do not exploit the underlying structure of ranked data.

Ranked data, despite being high dimensional, is structured, and being able to compute similarities suffices in many applications. For example, in recommender systems, the similarity of two users based on their preferences is vital to propose algorithms that provide recommendations to them. For this reason, kernels, which can be viewed as measures of similarities, are particularly well suited to model ranked data.

The first application of kernels to ranked data is the diffusion kernel (Kondor & Lafferty, 2002). However, the diffusion kernel is prohibitively expensive to compute in real-world settings as its time complexity scales

|  | Full | Top-$k$ | Exhaustive | Non-exhaustive |
|---|:---:|:---:|:---:|:---:|
| Mallows kernel | ✓ | ✓ | ✗ | ✗ |
| Kendall kernel | ✓ | ✓ | ✓ | ✗ |
| Graph cut kernel (ours) | ✓ | ✓ | ✓ | ✓ |

Table 1: Kernel for the different kinds of rankings data.

exponentially with the number of objects. Jiao & Vert (2015) propose the Kendall and Mallows kernels, which are based on the Kendall distance for rankings (Kendall, 1948), and a tractable algorithm to calculate them. These kernels correspond to the linear kernel and the squared exponential kernel on the symmetric group (Diaconis, 1988), which is the space yielded by interpreting rankings as permutations. For a set of $n$ objects, the time complexity for both kernels is $n \log n$ for full rankings. In the specific case of the Kendall kernel, the algorithm can be extended to partial rankings. Jiao & Vert (2018) propose a weighted version of the Kendall kernel to model the different contributions of the different pairs in the rankings but the $O(n \log n)$ time complexity algorithm is only valid for full and top-$k$ rankings. Finally, Lomeli et al. (2019) derive a Monte Carlo kernel estimator for the Mallows kernel for the specific case of top-$k$ rankings. The main challenge of applying kernel methods to ranked data is to efficiently deal with all kinds of rankings: full, top-$k$ and interleaving. Previous kernels for ranked data (Jiao & Vert, 2015; Lomeli et al., 2019) have been tailored for specific types of rankings; see Table 1 for an overview of the state of the art.

In this paper, we propose the graph cut kernel, which can attain good performance at a low computation budget *regardless* of the specific kind of ranking. To achieve this challenging objective, we interpret ranked data in terms of ordered partitions, rather than permutations on the symmetric group. This allows us to use the efficient machinery of submodular functions. Submodular functions, which the graph cut function is an example of, have been previously applied to *ratings*[1] in the form of Lovász Bregman divergences (Iyer & Bilmes, 2013). In this paper, we propose a novel kernel for *rankings*. The main contribution of the paper is the graph cut kernel for all kinds of rankings (full, top-$k$, exhaustive interleaving and non-exhaustive interleaving). The graph cut kernel exploits the geometric structure of ranked data to drastically reduce the computational cost while still retaining good empirical performance.

The structure of the article is as follows. In Section 2, we show how to encode all kinds of rankings as ordered partitions and their relationship with submodular functions, in particular with the graph cut function. In Section 3, we propose a *monotonic* feature map using the submodular functions that is consistent with the ranking of the objects. Finally, in Section 4, we support our claims and demonstrate scalability empirically using synthetic and a real datasets.

## 2  Ordered partitions and submodular functions

In Section 2.1, we introduce the different types of rankings and how to encode them as ordered partitions and, in Section 2.2, we revise submodular functions. Further, in Section 2.3, we introduce the base polytope, a geometrical representation of the space of ordered partitions, which is yielded by submodular functions and is characterised by an exponential growth of the number constrains with the number of objects. Section 2.5 presents the tangent cone, an outer approximation of the base polytope, which is instead characterised by constraints that scale linearly in the number of objects. This cheap but representative approximation for rankings will be used to derive the feature map of the graph cut kernel in Section 3. It turns out that the monotonicity of the feature map is consistent with the ranking of the objects.

### 2.1  Ranked data and ordered partitions

Ranked data is often available in different forms. Table 2 contains the most common kinds of rankings objects, where $a \prec b$ denotes the preference of $b$ over $a$. In full rankings, the pairwise ordering between

---

[1]Ratings differ from rankings in the way the preference is expressed. Ratings are defined using a scale of preference, whereas rankings only encode the order of preferences.

Table 2: Examples of rankings from a set of 5 objects.

| TYPE | RANKING | ORDERED PARTITION |
|------|---------|-------------------|
| Full | $2 \prec 1 \prec 3 \prec 4 \prec 5$ | $\{2\} \prec \{1\} \prec \{3\} \prec \{4\} \prec \{5\}$ |
| Top-2 | $2 \prec 3$ | $\{4,5,1\} \prec \{2\} \prec \{3\}$ |
| Exh. | $1 \prec 2 \prec 3 \cap 1 \prec 2 \prec 4 \cap$ | $\{1\} \prec \{2,5\} \prec \{3,4\}$ |
| | $1 \prec 5 \prec 3 \cap 1 \prec 5 \prec 4$ | |
| Non-exh. | $2 \prec 1 \prec 4$ | $\{2\} \prec \{1\} \prec \{4\}$ |

all objects is known. In top-$k$ rankings, the information on the pairwise ordering is only available for the $k$ most favourite objects. This implicitly means that the order of the remaining objects is unknown. For exhaustive rankings, we know the exact ordering of some subsets of the items but not the pairwise ordering of the items inside the subsets. For example, given $\{1,2\} \prec \{3,4\}$, we know that both 3 and 4 are preferred to 1 and 2 but we lack the information about the exact ordering of 3 compared to 4 and 1 compared to 2. Non-exhaustive rankings only contain information on a subset of objects and, therefore, the ordering of the other objects is unknown. For instance, in the example of non-exhaustive ranking given in Table 2, object 3 may lie between object 2 and object 1 or object 1 and object 4.

In this paper, we encode rankings using *ordered partitions* to achieve a compact and general representation. Let $\mathcal{V}$ be the set of objects, then an ordered partition can be represented by $l$ mutually exclusive sets $A_1, \ldots, A_l$, such that

$$A_1 \prec \ldots \prec A_l. \tag{1}$$

If the subsets cover all objects, i.e. $\cup_{i=1}^{l} A_i = \mathcal{V}$, then we refer to this as an *exhaustive* ordered partition, otherwise as a *non-exhaustive* ordered partitions. Full and top-$k$ rankings are special cases of exhaustive ordered partition. However, non-exhaustive rankings can only be represented by non-exhaustive partial ordering.

In the following, we introduce submodular functions and a specific polytope related to submodular functions whose faces are characterized by ordered partitions. In Section 3, we then define a monotonic feature map that is consistent with a ranking that is represented by an ordered partition.

## 2.2 Submodular functions

Submodular functions are set functions characterized by diminishing returns, which makes them suitable for many applications ranging from game theory to electrical networks. Formally, let $n$ be the number of objects in set $\mathcal{V}$. A set function defined on the power set $F : 2^{\mathcal{V}} \to \mathbb{R}$ is submodular if, for all subsets $A_i, A_j \subset \mathcal{V}$

$$F(A_i) + F(A_j) \geq F(A_i \cup A_j) + F(A_i \cap A_j). \tag{2}$$

The power set $2^{\mathcal{V}}$ is naturally identified by the vertices $\{0, 1\}^n$ of an $n$-dimensional unit hypercube. Therefore, a set function $F : 2^{\mathcal{V}} \to \mathbb{R}$ defined on the power set of $\mathcal{V}$ is the same as defining it on the vertices of the $n$-dimensional unit hypercube. Each set function may be extended to the complete hypercube $[0, 1]^n$ using the Lovász extension (Lovász, 1982), which we refer to as $f : [0, 1]^n \to \mathbb{R}$. For any vector $w \in [0, 1]^n$, we may calculate the value of the function $f(w)$ and find the permutation of objects $j_1, \ldots, j_n$, such that $w_{j_1} \geq \ldots \geq w_{j_n}$. Then

$$f(w) = w_{j_n} F(\{j_1, \ldots, j_n\}) + \sum_{k=1}^{n-1} (w_{j_k} - w_{j_{k+1}}) F(\{j_1, \ldots, j_k\}). \tag{3}$$

For example, let us take the set $\mathcal{V} = \{a, b\}$ with power set $2^{\mathcal{V}} = \{\varnothing, \{a\}, \{b\}, \{a, b\}\}$. Each subset may be represented by an indicator vector of length $|\mathcal{V}| = 2$. The first index of the indicator vector of the subset represents the presence of the object $\{a\}$ in it while the second index represents the presence of the object $\{b\}$. Therefore, the corresponding indicator vectors $\{[0,0]^T, [1,0]^T, [0,1]^T, [1,1]^T\}$ form the corners of the

two-dimensional unit hypercube $[0,1]^2$. If $w = \begin{pmatrix} w_a \\ w_b \end{pmatrix}$, then the Lovász extension $f$ is

$$f(w) = \begin{cases} (w_b - w_a)F(\{b\}) + w_a F(\{a,b\}) & \text{if } w_b \geq w_a, \\ (w_a - w_b)F(\{a\}) + w_b F(\{a,b\}) & \text{if } w_a \geq w_b. \end{cases} \tag{4}$$

This extension is piecewise linear for any set function $F$, and it is tight at the corners of the hypercube. Furthermore, $f$ is convex if and only if the corresponding set function $F$ is submodular (Lovász, 1982). The Lovász extension may be defined on $\mathbb{R}^n$ using the same notion.

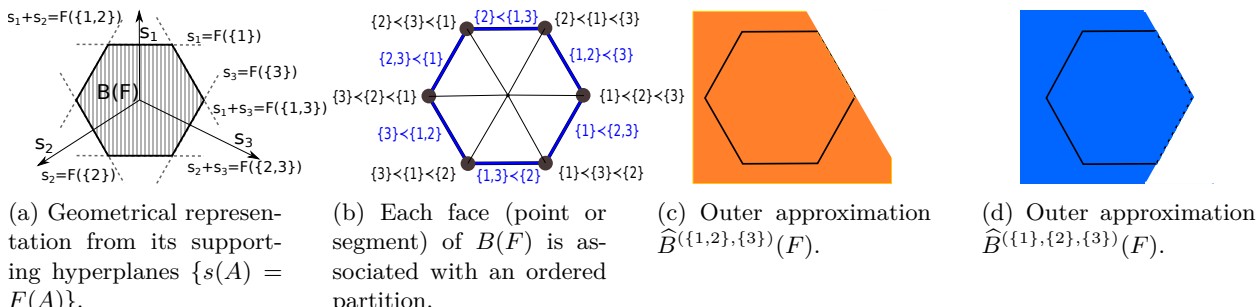

(a) Geometrical representation from its supporting hyperplanes $\{s(A) = F(A)\}$.

(b) Each face (point or segment) of $B(F)$ is associated with an ordered partition.

(c) Outer approximation $\widehat{B}^{(\{1,2\},\{3\})}(F)$.

(d) Outer approximation $\widehat{B}^{(\{1\},\{2\},\{3\})}(F)$.

Figure 1: Overview of the base polytope and its outer approximations for $n=3$.

## 2.3 Base polytope

Any piecewise linear convex function may be represented as the support function $f(w) = \max_{s \in K} \langle w, s \rangle$ of a certain polytope $K$ (Rockafellar, 1997). For the Lovász extension of a submodular function, there is an explicit description of $K$, known as the *base polytope*, which we now describe.

**Definition 2.1** *Base polytope.* Let $s(B) = \sum_{i \in B} s_i$, where $s_i$ is the $i^{th}$ element of $s \in \mathbb{R}^n$. The base polytope

$$B(F) := \left\{ s(B) \leq F(B) \cap s(\mathcal{V}) = F(\mathcal{V}) \,|\, \forall B \subset \mathcal{V} \right\} \tag{5}$$

is included in the affine hyperplane $s(\mathcal{V}) = F(\mathcal{V})$. All the adjacent faces of the hyperplane in Figure 1a may be used to get a ranking in Figure 1b. A key result in submodular analysis (Fujishige, 2005) is that the Lovász extension is the support function of base polytope $B(F)$, i.e., for any $w \in \mathbb{R}^n$, we have

$$f(w) = \sup_{s \in B(F)} \langle w, s \rangle. \tag{6}$$

## 2.4 Graph cuts

In this paper, we are mainly interested in using a class of submodular functions called *graph cuts*. Let us assume we have a weighted undirected graph with the weights function $a : \mathcal{V} \times \mathcal{V} \to \mathbb{R}_+$. A graph cut is defined on any subset $A$ of the nodes of the graph $\mathcal{V}$ as

$$F(A) = \sum_{i < j} a(i,j)|1_{i \in A} - 1_{j \notin A}|,$$

where $i, j \in \mathcal{V}$. This is the sum of weights of all the edges of the graph that connect nodes in the set $A$ to the nodes that are not in set $A$, i.e., $\mathcal{V} \setminus A$. The corresponding Lovász extension is defined as

$$f(w) = \sum_{i < j} a(i,j)|w_i - w_j|,$$

where $w \in [0,1]^n$.

This function is of particular interest to this paper because we assume a value function oracle that may be evaluated in time linear in the number of edges of the graph. There are other rich classes of information-theoretic functions like entropy and mutual information that belong to the class of submodular functions. Although these functions can be successfully used to model several important problems, evaluating them is significantly expensive. This is one of the reasons we focus on graph cuts in this paper. Investigating applications that allow for efficient computation of such functions represent an interesting future direction.

### 2.5 Tangent cone

However, the number of constraints for the base polytope, Eq. (5), grows exponentially with the cardinality of $\mathcal{V}$, i.e. $2^n - 1$. For an ordered partition $A$, we can define the *tangent cone* of the base polytope $B(F)$, a form of outer approximation with constraints linear in the length of $A$, $l$.

**Definition 2.2** *Tangent cone.* Let $B_i := A_1 \cup \ldots \cup A_i$, which we also represent using $\cup_{j=1}^i A_j$ for brevity. We define the tangent cone of the base polytope $B(F)$ characterized by the ordered partition $A$ as

$$\widehat{B}^A(F) = \left\{ s(B_i) \leq F(B_i) \cap s(\mathcal{V}) = F(\mathcal{V}), \forall i \in \{1, \ldots, l-1\} \right\}. \tag{7}$$

The tangent cone considers only $l - 1$ constraints of the exponentially many constraints used to define the base polytope, and it forms an outer approximation of the ordered partitions as illustrated in Figure 1c and d for $\big(\{1, 2\} \prec \{3\}\big)$ and $\big(\{1\} \prec \{2\} \prec \{3\}\big)$, respectively.

Since the number of constraints of the tangent cone is *linear* in the length of the ordered partition, it is possible to leverage efficient submodular optimization techniques to retrieve a cheap approximation for ranked data using their tangent cone representations. The graph cut kernel is built on top of this approximation as explained in Section 3.1.

## 3 Kernel for ordered data

Using the connections between ordered partitions and submodularity (Section 3.1) we propose a feature map for the graph cut kernel. Section 3.2 extends the kernel to non-exhaustive ordered partitions, i.e. non-exhaustive rankings.

### 3.1 From the feature map to the kernel

We propose to use the min-norm point on the tangent cone of the base polytope $\widehat{B}^A(F)$ as the feature map

$$\phi(A) := s^* = \operatorname*{arg\,min}_{s \in \widehat{B}^A(F)} \frac{1}{2} \|s\|^2 \in \mathbb{R}^n, \tag{8}$$

where $n$ is the number of objects. Kumar & Bach (2017) showed that this is dual to the *isotonic regression* problem, i.e., assuming that $B_0 = \emptyset$,

$$\min_{v \in \mathbb{R}^l} \sum_{i=1}^l v_i \big[ F(B_i) - F(B_{i-1}) \big] + \frac{1}{2} \sum_{i=1}^l |A_i| v_i^2$$
$$\text{s.t.} \quad v_1 \geqslant \cdots \geqslant v_l, \tag{9}$$

where the optimal solution $s^*$ of Eq. (8) and the optimal solution $v^*$ of Eq. (9) are related via

$$s^* = -\sum_{i=1}^l v_i^* 1_{A_i}. \tag{10}$$

We re-derive the details and show the pseudocode for the PAVA algorithm in Appendix A.

Note that $v_i^*$ corresponds to the set $A_i$, while $s_j^*$ corresponds to the object $j$ and $s_j^* = -v_i^*$, if object $j$ belongs to the subset $A_i$. The isotonic regression problem may be solved using the pool adjacent violator

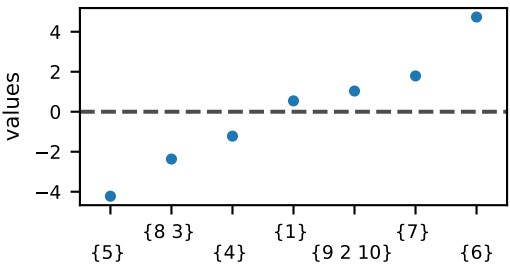

Figure 2: Optimal values $v^*$, Eq. (11), using the submodular cut function and the information graph of the sushi dataset for $A$.

algorithm (PAVA) (Best & Chakravarti, 1990). If the optimal solution $v^*$ has the same values for some of the partitions, i.e. $v_i^* = v_{i+1}^*$ for some $i \in \{1, \dots, l-1\}$, this corresponds to a merge of $A_i$ and $A_{i+1}$. If these merges are made, we obtain a *basic ordered partition*[2], such that our optimal $v^*$ has *strictly decreasing* values. Because none of the constraints are tight, primal stationarity leads to explicit values of $v^*$ given by

$$v_i^* = -(F(B_i) - F(B_{i-1}))/|A_i|, \tag{11}$$

i.e., the exact solution to the isotonic regression problem in Eq. (9) may be obtained in closed form.

We now present an example of a feature map for

$$A = 5 \prec 8 \prec 3 \prec 4 \prec 1 \prec 2 \prec 9 \prec 10 \prec 7 \prec 6$$

to support the choice of the feature map. Using $\phi$ allows us to derive a geometrically meaningful kernel.

Figure 2 reports an example of $v^*$ associated with $A$. Higher values correspond to *more preferred* objects and lower values to *less preferred* ones, whereas objects in the middle tend to be associated with values closer to zero. This particular structure of $v^*$ is characteristic of submodular functions $F$. The values $v^*$ are related to the feature map in Eq. (10) in the sense that each dimension $d$ of $\phi(A)$ corresponds to an object, and $\phi(A)_d$ is the value in $v^*$ corresponding to $d$. From the example in Figure 2, $\phi(A)_5 = -4.22$ , $\phi(A)_3 = \phi(A)_8 = -2.36$ and so on. Therefore, a natural and efficient way to exploit this pattern is to use the linear kernel on the resulting feature maps:

$$k_s(A, A') = \sum_{d=0}^{n} \phi(A)_d \phi(A')_d = \langle \phi(A), \phi(A') \rangle. \tag{12}$$

If in two rankings $A$ and $A'$, an object $d$ is ranked similarly then this results in the multiplication of two numbers with the same sign, i.e. $\phi(A)_d \phi(A')_d > 0$. On the contrary, if $d$ is ranked differently then the multiplication would happen between numbers with different signs and thus $\phi(A)_d \phi(A')_d < 0$. How much greater or less than zero depends on the position of the $d$ in the rankings. Figure 3 reports $\phi(A)$ and $\phi(A')$ where $A'$ is the inverse of $A$ along with the multiplication factor $\phi(A)_d \phi(A')_d$.

### 3.2 Kernel for non-exhaustive partial orderings

To extend the graph cut kernel from exhaustive to non-exhaustive ordered partitions, we recur to the convolution kernel (Haussler, 1999) by defining the set of exhaustive order partitions that are coherent with a non-exhaustive ranking, $R(\cdot)$. The convolution kernel over the graph cut kernel is

$$k_c(A, A') = \frac{1}{|R(A)||R(A')|} \sum_{a \in R(A)} \sum_{a' \in R(A')} k_s(a, a'), \tag{13}$$

---

[2]Given a submodular function $F$ and an ordered partition $A$, when the unique solution problem in Eq. (9) is such that $v_1 > \dots > v_l$, we say that we $A$ is a *basic ordered partition* for $F$. Given any ordered partition, isotonic regression allows to compute a coarser partition (obtained by partially merging some sets) which is basic.

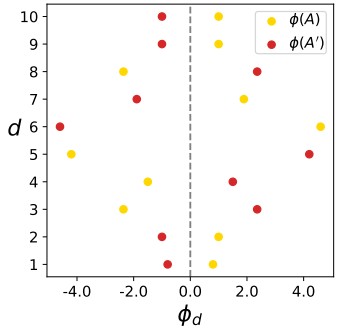
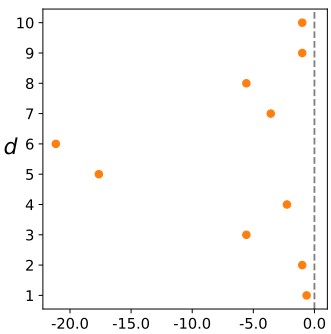

(a) Values for each dimension $d$ of the 10 dimensional feature maps $\phi(A)$ and $\phi(A')$.

(b) Multiplication values for each dimension of $\phi(A)$ and $\phi(A')$.

Figure 3: Geometrical interpretation of the linear kernel of feature map $\phi(A)$ and $\phi(A')$, where $A'$ is the inverse of $A$.

where $A, A'$ are non-exhaustive rankings ordered partitions and $R(A), R(A')$ are the set of exhaustive ordered partitions that are coherent with $A, A'$ respectively. Due to the linearity of the inner product, we can reduce $k_c(\cdot, \cdot)$ to the inner product between the average of the feature maps over $R(A)$ and $R(A')$, so that

$$
\begin{aligned}
k_c(A, A') =& \frac{1}{|R(A))||R(A'))|} \sum_{a \in R(A)} \sum_{a' \in R(A')} k_s(a, a') \\
=& \frac{1}{|R(A))||R(A'))|} \sum_{a \in R(A)} \sum_{a' \in R(A')} \langle \phi(a), \phi(a') \rangle \\
=& \Big\langle \frac{1}{|R(A)|} \sum_{a \in R(A)} \phi(a), \frac{1}{|R(A')|} \sum_{a' \in R(A')} \phi(a') \Big\rangle \\
=& \langle \tilde{\phi}(A), \tilde{\phi}(A') \rangle,
\end{aligned}
\tag{14}
$$

where $\tilde{\phi}(A)$ is the mean of the feature map over the set of exhaustive ordered partitions that are coherent with $A$. Similarly to $k_s$, we can pre-compute these feature maps.

The main drawback of using the convolution kernel for non-exhaustive rankings is that the time complexity for calculating the feature maps becomes exponential due to the exponential growth of $R(A)$ in the number of objects. Given an interleaving ranking $A$ of length $l$ for $n$ objects, $R(A)$ has cardinality $(l+1)^{n-l}$. In this case, we sample from the set of coherent ordered partitions and calculate the average feature map on the sampled subset of $R(A)$. This approximation allows us to scale well also for large sets of items. In Section 4, we show that we empirically obtain competitive results using this sampling procedure that removes the burden of the computational complexity.

### 3.3 The graph cut kernel

In summary, the algorithm to calculate the graph cut kernel happens in two sequential steps, (i) given a submodular function, we extract low-dimensional features from the data using PAVA, see Alg. 1, and (ii) we use the computationally inexpensive linear kernel. Assuming an oracle submodular function with time complexity $O(p)$ and an exhaustive ordered partition of length $l$ for $m$ samples, the time complexity for calculating the Gram matrix using the submodular kernel is $O(mpln + m^2 n)$. Note that these complexities hold only if all the rankings are exhaustive ordered partitions of at most size $l$. The time complexity for Gram matrix for rankings that are non-exhaustive is given by $O(mspln + m^2 n)$, where $s$ is the number of samples when we use sampling as explained in Section 3.2. In the next session, we empirically show that the graph cut kernel is, at the same time, able to match or outperform state-of-the-art performance in an established classification task for kernel-based methods.

---

**Algorithm 1** PAVA($\mathcal{A}$, F)

---

**Input:**
     - Ordered partition $\mathcal{A}$ of length $l$.
     - Submodular function $F$.
Initialize $v$. // Define a set to contain the optimal values
Initialize $B_1, \ldots, B_l$ // using Eq. (7)
$k \leftarrow l$.    // To retrieve sets $B_{k-1}$ and $B_k$
**while** $|v| < |A|$ **do**
    $v' \leftarrow -\frac{F(B_{k-1})-F(B_k)}{|B_{k-1}|-|B_k|}$ // optimal solution Eq. (11)
    **if** $v'$ violates the isotonic constraints, i.e. $v' > v_k$ **then**
      Merge partitions $A_k$ and $A_{k-1}$ and remove $B_k$.
    **else**
      $v_{k-1} \leftarrow v'$
      $k \leftarrow k - 1$
    **end**
**end while**
**Return** $v$

---

## 4 Experiments

The goal of this section is to demonstrate the (i) scalability, (ii) accuracy and (iii) flexibility of the graph cut kernel. We use a downstream classification task which is a traditional testbed for kernel-based methods for rankings (Jiao & Vert, 2015; Lomeli et al., 2019). The code is available at `https://github.com/MichelangeloConserva/CutFunctionKernel/`.

### 4.1 Datasets

The downstream classification task is performed on a synthetic dataset that we now describe and on an established real-world dataset.

**Synthetic dataset.** The synthetic dataset resembles food preferences of eight dishes

$$\{Cake, Biscuit, Gelato\}, \quad \{Pasta, Pizza\}, \quad \{Steak, Burger, Sausage\}.$$

partitioned in three groups based on the numerical features, which respectively describe them: *Sweetness*, *Savouriness* and *Juiciness*. The exact values for the features are reported in Table 3. There is a clear distinction between the first group and the others and a mild distinction between the second and the third one. Suppose that there are two types of users with opposite preferences $\{Sweet \prec Savouriness \prec Juicy\}$ (*type one* users) and $\{Juicy \prec Savouriness \prec Sweet\}$ (*type two* users), the classification task is to distinguish the types of users by their preferences.

| | cake | biscuit | gelato | steak | burger | sausage | pasta | pizza |
|---|---|---|---|---|---|---|---|---|
| sweet | 0.9 | 0.7 | 1.0 | 0.0 | 0.2 | 0.1 | 0.4 | 0.4 |
| savouriness | 0.0 | 0.1 | 0.0 | 0.8 | 0.8 | 1.0 | 0.7 | 0.9 |
| juicy | 0.3 | 0.0 | 0.7 | 0.8 | 0.9 | 1.0 | 0.7 | 0.6 |

Table 3: Numerical features for the objects of the synthetic dataset.

The main objective of this dataset is to reproduce a synthetic setting in which the features of the objects play an important role in determining the rankings. A probabilistic model that expresses such a setting is not available. We thus propose the following sampling procedure assuming that the two mentioned types of users rank the objects. We assume that both *type one* and *type two* users give different and identical importance

to the features of the objects when ranking them. We choose the weights for the most preferred feature to be 1, for the second 0.17 and for the last one 0.09 to reproduce an exponential decay in the preference. For example, *type one* presents a unitary importance weight for *Sweet*, a 0.17 importance weight for *Savouriness* and a 0.09 importance weight for *Juicy*. By summing the numerical features of the objects given in Table 3 using the importance weights of the two types of users we obtained two sets of scores for the objects, which Table 4 reports. From the table, it is easy to see that the scores given to the objects greatly vary for the two

|          | Cake | Biscuit | Gelato | Steak | Burger | Sausage | Pasta | Pizza |
|----------|------|---------|--------|-------|--------|---------|-------|-------|
| Type one | 0.93 | 0.72    | 1.06   | 0.21  | 0.42   | 0.36    | 0.58  | 0.6   |
| Type two | 0.38 | 0.08    | 0.79   | 0.93  | 1.05   | 1.18    | 0.85  | 0.79  |

Table 4: Scores given to the food objects by the two different types of users.

types of users and they have a strong connection with the underlying feature preferences.
In order to create a more challenging classification task, it is possible to inject Gaussian noise with $\sigma$ standard deviation to the scores and return the vector of indices sorted according to the jittered scores.
Formally, the sample procedure is,

$$\text{Ranking}_{\texttt{User1}} \sim \texttt{ArgSort}\big(\text{Score}_{\texttt{User1}} + \mathcal{N}(0, \sigma^2 I_8)\big) \in \mathbb{S}_8$$
$$\text{Ranking}_{\texttt{User2}} \sim \texttt{ArgSort}\big(\text{Score}_{\texttt{User2}} + \mathcal{N}(0, \sigma^2 I_8)\big) \in \mathbb{S}_8,$$

where $\mathbb{S}_8$ is the permutation space yielded by the eight objects in the dataset, $\texttt{ArgSort}$ returns the indices the list in input and $\text{Score}_{\texttt{User}}$ refers to the scores as in Table 4.
In Table 5 we report the underlying noise-free preferences of the two types of users which can be derived from Table 4. The preferences are expressed in increasing order, i.e. zero is associated with the least favourite object and 7 to the most preferred.

|          | Cake | Biscuit | Gelato | Steak | Burger | Sausage | Pasta | Pizza |
|----------|------|---------|--------|-------|--------|---------|-------|-------|
| Type one | 6    | 5       | 7      | 0     | 2      | 1       | 3     | 4     |
| Type two | 1    | 0       | 2      | 5     | 6      | 7       | 4     | 3     |

Table 5: Noise-free objects preferences for the two types of users. Higher values correspond to more preferred objects.

**Real-world dataset.** The state-of-the-art dataset for rankings data is the sushi dataset (Kamishima, 2003). Japanese users were asked to rank two different sets of sushis. In the first set, there are 10 sushis and the users express full rankings whereas in the second set there are one hundred sushis and the users express non-exhaustive rankings of ten of them. We refer to them as *small* and *large* sushi datasets. The preference in sushis varies between the two parts of the country so the origin of the person can meaningfully be the label for a classification task (Kamishima, 2003).

## 4.2 Classification

To assess the generalization capability of the graph cut kernel and to compare it with the state-of-the-art kernels, we fit Gaussian process (GP) classifiers. For the synthetic dataset, we choose a sample size of 250 for full and top-$k$ rankings and, due to the computational cost of the Kendall kernel, 100 for exhaustive interleaving rankings. For both the small and the large sushi datasets, we use 2500 rankings.

**Experimental setting.** Samples are split into train 80% and test sets 20%. Results are reported in terms of F1-scores (Chinchor, 1992), which provides a reliable estimate of the actual capabilities of the model as it penalizes the classifier for false positives and false negatives. Note that for a perfectly balanced dataset, i.e. equal number of samples for each category, the F1-score is equal to the accuracy score. To train the

GP classifier we use the python package GPflow (Matthews et al., 2017), which we extended with custom classes. The cut function is applied to a graph built from the information on the objects, i.e., the dishes in the synthetic dataset and the sushis in the real dataset. The edges of such graph are calculated using the squared exponential kernel as the similarity score function, i.e.,

$$e(v, v') = \exp\left(-\frac{1}{2}\frac{\|v - v'\|_2^2}{\ell^2}\right). \tag{15}$$

The lengthscale $\ell$ is selected using the inverse median heuristic (Schölkopf & Smola, 2002). For the large sushi dataset, we use 600 samples from the set of exhaustive ordered partitions $R$ and we retain the top 10% edges of the graph.

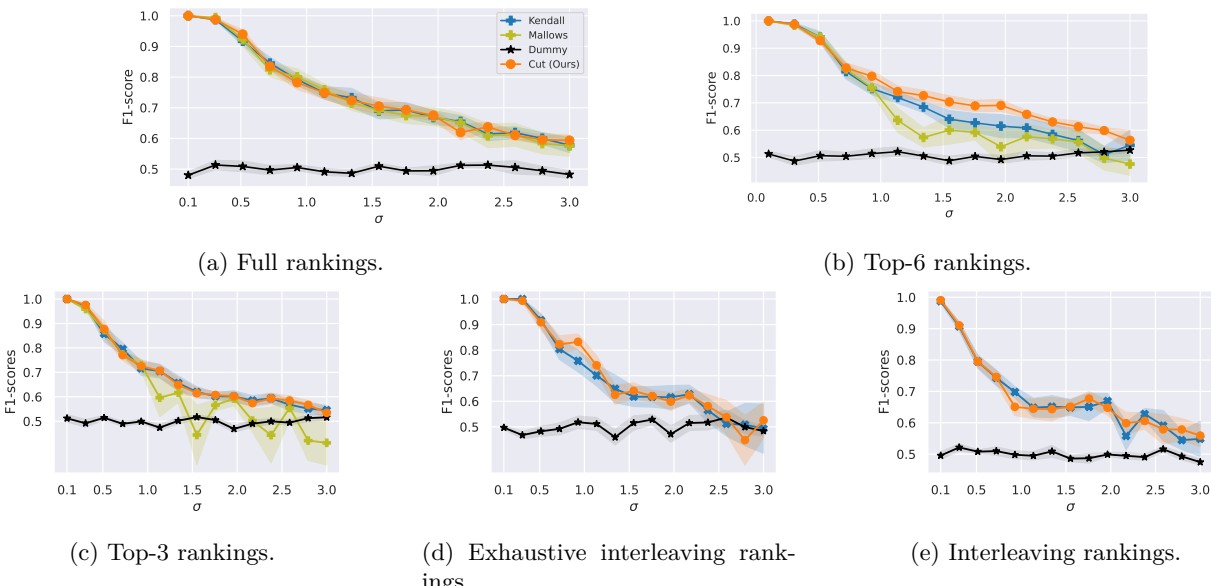

(a) Full rankings.      (b) Top-6 rankings.

(c) Top-3 rankings.      (d) Exhaustive interleaving rankings.      (e) Interleaving rankings.

Figure 4: We perform a train/test procedure on the synthetic dataset using 250 samples for each increasing level of noise and all the different kinds of rankings. We report the F1-scores calculated on test data. generated from the synthetic dataset. We average the results over six seeds.

**Synthetic dataset.** The synthetic dataset aims to show the ability of the kernels to handle noisy data. Figure 4 reports the F1-scores on the test sets obtained by the GP classifiers for the different kernels on increasing levels of noise. The graph cut kernel performs in line with state-of-the-art kernels and, for the specific case of top-6 rankings, Figure 4b, slightly better. We note that the performance of graph cut kernel for full and for the top-6 rankings matches. This shows that, given a mild level of censoring as in the top-6 scenario, the graph cut function on the information graph can successfully reconstruct the underlying preferences.The Mallows kernel exhibits a sharp drop in performance in the top-6 scenario and significantly worse performance in the top-3 one, Figure 4b. For both the exhaustive (Figure 4d) and non-exhaustive interleaving rankings (Figure 4e), the graph cut kernel matches the performance of the state-of-the-art Kendall kernel. The last two mentioned scenarios present a higher level of variability in performance. For the exhaustive interleaving rankings scenario, this is caused by the small sample size of 100, whereas for interleaving rankings, the variability is due to the task difficulty.

**Real-world dataset.** The sushi dataset is becoming the benchmark dataset to show the generalization capabilities of kernel methods on rankings, (Kondor & Lafferty, 2002; Lomeli et al., 2019). For this challenging classification task the performance on the test set, regardless of the kernel, is limited to an F1-score of 0.6. Nonetheless, our graph cut kernel slightly outperforms the state-of-the-art kernels. More specifically, in the full rankings scenario, Figure 5a, the F1-scores achieved by the graph cut kernel is slightly better compared to the Mallows kernel and better than the Kendall kernel. For the case of the top-6 ranking, Figure 5b, the

graph cut kernel is the only kernel to achieve better than random performance. The case of interleaving rankings presents a challenge both for the Kendall and the graph cut kernel, Figure 5c and Figure 5d. We note that the performance of the graph cut kernel can still be considered superior to the Kendall kernel as it is less variable across different seeds and slightly better in the large dataset, Figure 5d.

Using synthetic and real datasets we demonstrated that the graph cut kernel works at least as well as state-of-the-art kernels used for ranking.

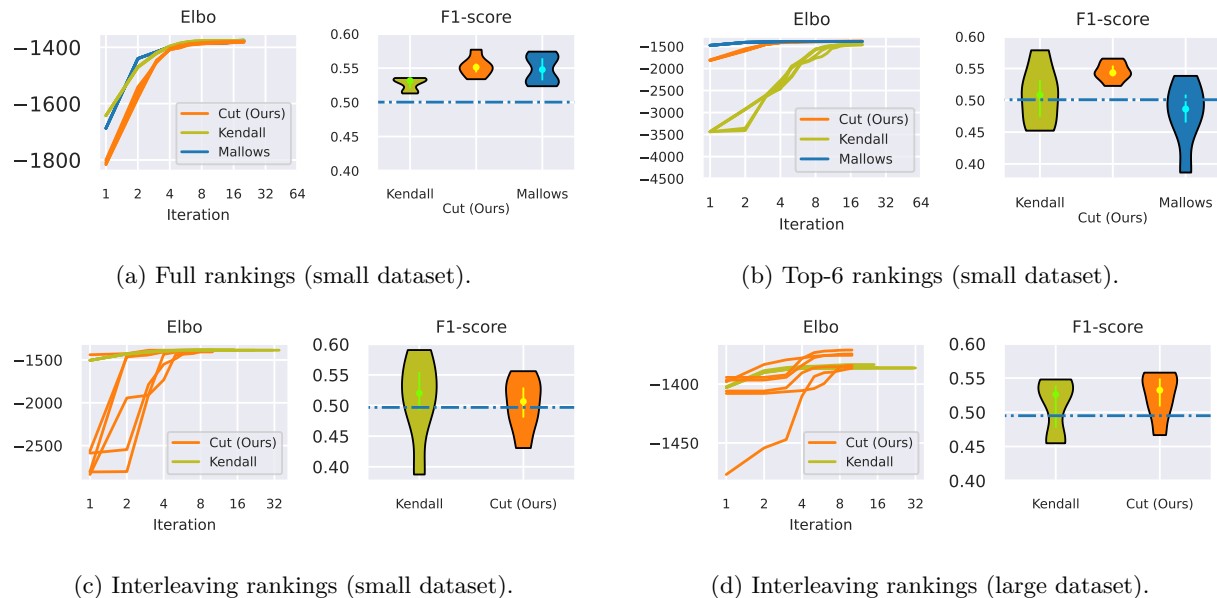

(a) Full rankings (small dataset).

(b) Top-6 rankings (small dataset).

(c) Interleaving rankings (small dataset).

(d) Interleaving rankings (large dataset).

Figure 5: Train/test procedure for GP repeated over six seeds for the sushi dataset. The small sushi dataset is employed for full (a), top-6 (b) and exhaustive rankings (c), whereas the large sushi dataset is used for non-exhaustive rankings (d). For each kind of ranking, we report the evidence lower bound (ELBO) during training (*Left*) and the F1-scores calculated on the test sets with a uniform dummy classifier baseline (*Right*). Note that the $x$-axis for the ELBO plot is in logarithmic scale and the mean, first quartile and third quartile are reported in bright colours in the violin plots.

### 4.3 Empirical time complexity

In the following, we compare the empirical time complexity for all kernels and highlight the steep advantage of the graph cut kernel. For algorithms meant to be employed in real-world applications, it is essential to analyse the empirical time complexity of computing the Gram matrix, in addition to the theoretical one that was covered in Section 3. In this section, we provide this empirical analysis using rankings sampled from the synthetic dataset.

**Experimental setting.** The values for the empirical time complexity are averaged over seven trials on an Intel Core i7-7700HQ CPU @ 2.80GHz. Differently from state-of-the-art methods, the computation of the graph cut kernel can make use of GPU hardware acceleration. However, to present a fair comparison, we limit the graph cut kernel calculation to CPU only. The accessibility of hardware acceleration for the graph cut kernel is easily accessible using Python packages such as GPFlow. In other words, after obtaining the feature maps for the rankings, the user can directly feed them into GPFlow that takes care of the GPU acceleration and additional optimizations. To give an idea of the scale of such optimization we report the total time required to calculate the Gram matrix for the entire dataset, i.e. 5000 samples. We found that the time required to calculate the feature maps is $6.31s \pm 88.6ms$, while the time required to compute the Gram matrix from the feature maps is $576ms \pm 19.7ms$ with hardware acceleration using a single Nvidia

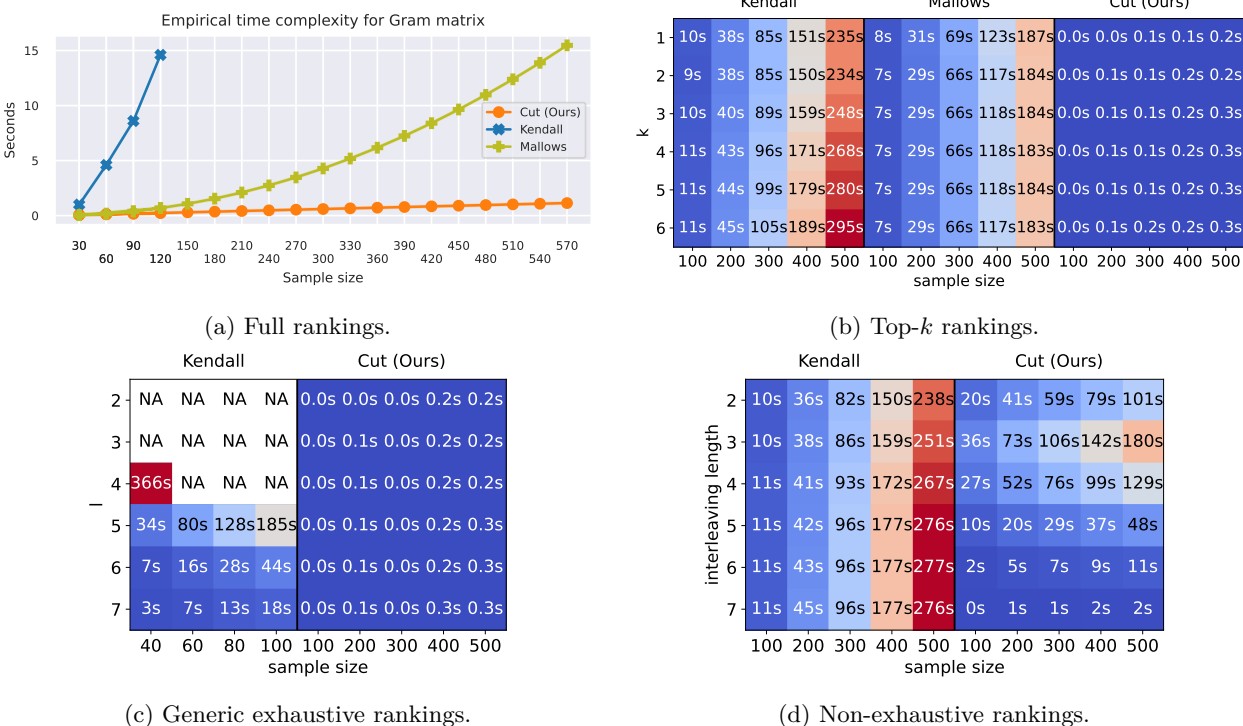

Figure 6: Comparison of empirical time complexity for calculating the Gram matrix using the submodular kernel and state-of-the-art kernels. Each figure reports the results for a different kind of ranking. Note that the symbol *NA* is reported when the empirical time complexity exceeds the maximum allowed time per simulation (seven minutes).

GeForce GTX 1060 GPU. Considering that the calculation of the feature maps does not happen when fitting the model, it is clear that the graph cut kernel can be confidently employed for large-scale datasets.

**Experiments.** Figure 6 reports the time spent for computing the Gram matrix as a function of the sample size and, for partial rankings, also of the level of censoring. For all kinds of exhaustive interleaving rankings (Figure 6a, b and c) computing the Gram matrix of the graph cut kernel is extremely cheap compared to state-of-the-art kernels. For the interleaving case, Figure 6d, due to the exponential factor, the time required is high but still significantly lower than the time required to compute the Gram matrix using the Kendall kernel. In Figure 6c some values for the Kendall kernel are not reported. This is because the Kendall kernel becomes computationally intractable in practice when a high number of ties are present in the rankings as it happens for low values of exhaustive interleaving ranking length. The massive speed advantage of the graph cut kernel comes from the fact that the computationally intensive calculations are done once per feature map and that the space complexity of storing the feature maps is linear both in the sample size and in the number of object[3]. In practice, the graph cut kernel scales almost linearly with the sample size.

## 5 Discussion

The graph cut kernel is a theoretically founded kernel that builds on top of the efficient machinery of submodular optimization. Thanks to this efficiency, we have shown a drastic reduction in the computational cost compared to previous kernel-based methods for ranked data. Additionally, it inherits all the theoretical properties of the linear kernel since it is built on top of it.

---

[3]The time for calculating the feature maps is included in the presented empirical time complexity for the graph cut kernel.

Permutohedrons are discrete objects best suitable for encoding ranked data. However, they have exponentially many vertices where each vertex corresponds to a permutation. This property makes the computation of the feature representation complex. In our approach, we use the graph cut function that shows a principled way to represent the permutohedron through its base polytope. These functions are designed using additional application-specific information. This provides us with a principled approach to traverse through the base polytope to provide an efficient feature representation. We have shown this empirically through several experiments.

Graph cuts are submodular functions and it is only this property that we have used to design the kernels. We believe general submodular functions specific to an application (e.g. entropy, mutual information etc.) may be used to design such kernels. This does not require any changes from our approach apart from designing submodular functions and using application-specific data.

## 6 Conclusion

In this paper, we presented a theoretically founded methodology to apply submodular function optimization for ranked data of all kinds: full, top-$k$ and interleaving. This allowed us to define the graph cut kernel that attains good empirical performance while only incurring fairly small computational costs. Thanks to the graph cut kernel, we enabled the use of kernel-based methods for large-scale datasets of rankings with the hope of encouraging a more widespread use of kernel methods for rankings. We support our claims by empirically training Gaussian process classifiers using both synthetic and real datasets and reporting the empirical time complexity. The graph cut kernel has proved to match or outperform the performance of the state-of-the-art kernels both in terms of accuracy on the classification task and of clock time.

We believe that the graph cut kernel will greatly benefit both practitioners, as they now have access to a wider range of tools for real-world applications, and theoreticians, as we have consolidated the theoretical connection between the graph cut function optimization and ranked data. We have thus opened several exciting research directions. We leave as an open problem the extension of the graph cut kernel to general submodular functions, which will require a different way to extract information from the rankings. Another possible future research direction is to adapt the graph cut kernel from ranking to ratings.

### Acknowledgments

The work is supported by Data Science Institute, Imperial College London.

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

## A Duality between isotonic regression and min-norm-point on the tangent cone

Let us recall that $F : 2^{\mathcal{V}} \to \mathrm{b}_+$ is a submodular functions and let $f : \mathrm{b}^n \to \mathrm{b}_+$ be its Lovász extension. We represent the *base polytope* of $F$ by $B(F)$. Given an ordered partition $A = (A_1, \ldots, A_l)$ of the set of objects $\mathcal{V}$, we recall the definition of tangent cone of the base polytope and derive its support function. Note that it is already derived by Kumar & Bach (2017) that we re-derive here for completion.

### A.1 Tangent cone and its support function.

Let $B_i := \cup_{j=1}^i A_j$. We define the tangent cone of the base polytope $B(F)$ characterized by the ordered partition $A$ as

$$\widehat{B}^A(F) = \Big\{ s(B_i) \leq F(B_i) \cap s(\mathcal{V}) = F(\mathcal{V}), \forall i \in \{1, \ldots, l-1\} \Big\}. \tag{16}$$

We can now proceed to compute the support function of the tangent cone $\widehat{B}^A(F)$, which is an upper bound on $f(w)$ since this set is an outer approximation of $B(F)$:

$$\sup_{s \in \widehat{B}^A(F)} w^\top s = \sup_{s \in \mathrm{b}^n} \inf_{\lambda \in \mathrm{b}_+^{l-1} \times \mathrm{b}} w^\top s - \sum_{i=1}^m \lambda_i(s(B_i) - F(B_i)) \text{ , using Lagrangian duality,} \tag{17}$$

$$= \inf_{\lambda \in \mathrm{b}_+^{l-1} \times \mathrm{b}} \sup_{s \in \mathrm{b}^n} s^\top \Big( w - \sum_{i=1}^l (\lambda_i + \cdots + \lambda_l) 1_{A_i} \Big) + \sum_{i=1}^l (\lambda_i + \cdots + \lambda_l) \big[ F(B_i) - F(B_{i-1}) \big], \tag{18}$$

$$= \inf_{\lambda \in \mathrm{b}_+^{l-1} \times \mathrm{b}} \sum_{i=1}^l (\lambda_i + \cdots + \lambda_l) \big[ F(B_i) - F(B_{i-1}) \big] \tag{19}$$

$$\text{such that } w = \sum_{i=1}^l (\lambda_i + \cdots + \lambda_l) 1_{A_i}. \tag{20}$$

Thus, by defining $v_i = \lambda_i + \cdots + \lambda_l$, which are decreasing, the support function is finite for $w$ having ordered level sets corresponding to the ordered partition $A$ (we then say that $w$ is *compatible* with $A$). In other words, if $w = \sum_{i=1}^l v_i 1_{A_i}$, the support functions is equal to the Lovász extension $f(w)$. Otherwise, when $w$ is not compatible with $A$, the support function is infinite.

Let us now denote $\mathcal{W}^A$ as a set of all weight vectors $w$ that are compatible with the ordered partition $A$. This can be defined as

$$\mathcal{W}^A = \Big\{ w \in \mathrm{b}^n \mid \exists v \in \mathrm{b}^l, w = \sum_{i=1}^l v_i 1_{A_i}, v_1 \geq \ldots \geq v_l \Big\}. \tag{21}$$

Therefore,

$$\sup_{s \in \widehat{B}^A(F)} w^\top s = \begin{cases} f(w) & \text{if } w \in \mathcal{W}^A, \\ \infty & \text{otherwise.} \end{cases} \tag{22}$$

### A.2 Duality

For all $w \in \mathcal{W}^A$, i.e., $w = \sum_{i=1}^l v_i 1_{A_i}, v_1 \geq \ldots \geq v_l$.

$$f(w) + \tfrac{1}{2} \|w\|_2^2 = \sum_{i=1}^l v_i \big[ F(B_i) - F(B_{i-1}) \big] + \tfrac{1}{2} \sum_{i=1}^l |A_i| v_i^2. \tag{23}$$

Let us now consider the minimization of the above optimization problem. It can be rewritten as a weighted isotonic regression problem of the form

$$\min_{\alpha \in \mathrm{b}^l, \alpha_1 \leq \cdots \leq \alpha_n} \sum_{i=1}^l \beta_i (\alpha_i - a_i)^2, \tag{24}$$

where $a \in \mathrm{b}^l$ and $\beta \in \mathrm{b}^l_+$ are given. Note the change of direction of monotonicity from $v$ to $\alpha$ for a standard isotonic regression setting. We now derive the dual of the above problem as

$$\min_{w \in \mathcal{W}^A} f(w) + \tfrac{1}{2}\|w\|_2^2 \overset{Eq. \ (22)}{=} \min_{w \in \mathrm{b}^n} \max_{s \in B^A(F)} s^\top w + \tfrac{1}{2}\|w\|_2^2 \tag{25}$$

$$= \max_{s \in B^A(F)} \min_{w \in \mathrm{b}^n} s^\top w + \tfrac{1}{2}\|w\|_2^2 = \max_{s \in B^A(F)} -\tfrac{1}{2}\|s\|_2^2, \tag{26}$$

where $s^* = -w^*$ at optimal and the dual problem, $\max_{s \in B^A(F)} -\tfrac{1}{2}\|s\|_2^2$ is to find the min-norm-point on the tangent cone of the base polytope.

The isotonic regression problem of the form Eq. (24) may be optimized using the PAVA algorithm proposed by Best & Chakravarti (1990) in time $O(ln)$ time complexity. This has also been used in the context of submodular function minimization (Appendix A.3 of Bach (2013); Kumar & Bach (2017)).

