# OpenReview forum: "The Graph Cut Kernel for Ranked Data"
_TMLR — Accepted by TMLR_

### Review · Reviewer_moz1 · 2022-04-20

**Summary Of Contributions:**

The paper proposes making use of min-norm point of the tangent cone of the base polytope as the feature embedding space to evaluate similarity amongst rankings. The entire machinery for the setup is already present in previous works. This paper's contribution is identifying this feature embedding space as a potential similarity embedding metric, and presents a few experiments to justify the use of proposed kernel space, in terms of classification accuracy and speedup.

**Requested Changes:**

Please see the weakness section for more detailed discussion. The main points would be a more thorough empirical investigation, cleaning up the writing, better motivation of the use of the said kernel especially for partial rankings.

**Strengths And Weaknesses:**

Strengths: Background is adequately discussed, and the idea itself has good potential.

Weaknesses:

I would classify this paper as somewhat of a “methods” paper i.e. the paper proposes a new similarity metric that to aid in relatively fast similarity evaluations based on full or partial order rankings. While the paper does not propose a new algorithm, the proposed kernel, especially because of its ease of computation could be impactful in several problems. However, given that the potential usefulness or impact isn’t analyzed theoretically (with guarantees etc.), the empirical section is less than convincing. I would disagree with the authors’ assertion on the sushi dataset being “state of the art” for evaluation. Is there a reason not to extract out partial rankings from user-movie datasets, for example, and do a more thorough evaluation? The currently presented experiments are hardly “proof of concept”, rather than being a useful indicator of applicability of the proposed kernel.

The paper also lacks motivation for using the said feature embedding as appropriate for measuring ranking similarity. There is an example that the authors take for a size 10 full ranking order, but it is unclear how the absolute value of \phi_d is indicative of similarity, especially for the case when the order is not full but only partial. Expanding on this will be helpful.

This is especially aggravated by the fact that the partial rankings are handled by mere averaging. Moreover, to speed up large computations, random subsampling is proposed which leads to large variances in performance. Further concentrated efforts are required to assert if the proposed kernel is practically viable.

One way to expand on the empirical side would be to use ratings datasets, as mentioned above. It is unclear why is this not straightforward and the authors have left it as future work in their conclusion? As most ratings are arbitrary anyways, and what really matters is preferences in users. Why then can the authors not extract preference information from ratings and use these preferences as partial orders for their framework? This will help strengthen the otherwise weak empirical section. Currently the propose method seems to perform reasonably well compared to baselines on full rankings, but the performance on the tradeoff between speed vs accuracy has large variances for the limited experimental evaluation on a single real-world dataset that is presented. The timing experiments are not suprising, given how the computations are performed.

The authors state in the conclusion -- “as we have consolidated the theoretical connection between the graph cut function optimization and ranked data “. I think this is an over-sell, the base polytope is already well-known, so is the lovasz extension that is defined on partitions governed by rankings.

Regarding “Extension to general submodular functions” – Can the authors please clarify this a bit more? What would such use-cases look-like ? An example or two would be very helpful.


Minor clarifications/Typos:

``and it is traditionally represented by the projection onto that hyperplane” in Def 2.1 is unclear. While the mathematical formulations are clear, the statement made for the projection onto the hyperplane does not make sense to me. In the same vain, Fig 1(b) could use more discussion. I would suggest a 3d image for Fig 1(a) to clarify the implication on the ranking shown in Fig 1(b).

Def 2.2: Please define A within the definition before saying B = \cup_i A_i. What is “l” and where was it defined ? Please write it as size of A explicitly.

Why is it called “graph cut kernel” ?
Clarify -- “Note that vi∗ corresponds to the set Ai, while s∗j corresponds to the object j and s∗j = −vi∗, if object j
belongs to the subset Ai. “
Indices in eq 18 seem to be incorrect.
What is ‘m’ in eq 17? Is it supposed to be ‘l’ ?
Please label the dots in Fig 3b with respective indices. There are 9 dots because d=9,10 result in the same similarity value ?
“which is yield by submodular functions “

“ exponential number of constrains in the number of objects. “

Sec 4.1 for the synthetic dataset, the ordering mentioned to identify type one and type two users is exactly the same. Please fix to differentiate.

---

> ### Author Response · Authors · 2022-05-16
> **Contribution**
>
> > I would classify this paper as somewhat of a “methods” paper i.e. the paper proposes a new similarity metric that to aid in relatively fast similarity evaluations based on full or partial order rankings.
>
> The state of the art kernels do not use any form of structural information from the dataset. The reason we improve computationally considerably is because we have been able to use additional information that can be encapsulated in the form of Graphs which in turn helps us design efficient kernels. This approach has not been attempted earlier with kernels for ranked data.
>
> > The paper also lacks motivation for using the said feature embedding as appropriate for measuring ranking similarity.
>
> The strictly increasing nature of the feature map is mainly due to the use of submodular functions. Let us consider two partial rankings $1 < 4$ and $1 < 5 < 4$. The values of the feature map for the object $4$ will certainly be higher in the later situation when compared to earlier because the averaging pushes the value down. We do agree that the computational gain pays in terms of variance theoretically. However, we have achieved similar or better results than the state of the art algorithms even with this variance for the applications shown.
>
> > Extension to general submodular functions.
>
> We have favour the graph cut function because, contrary to other submodular function such as the entropy or the mutual information, it can be computed efficiently. We have introduced section 2.4 on Graph cuts and why we favour graph cuts over general submodular functions.
>
> > the empirical section is less than convincing. I would disagree with the authors’ assertion on the sushi dataset being “state of the art” for evaluation. Is there a reason not to extract out partial rankings from user-movie datasets, for example, and do a more thorough evaluation?
>
> The large version of the sushi dataset (5000 samples and 100 items) is the large scale dataset we have used (see Fig. 5(d)). We did not employ other large scale datasets because the only state-of-the-art kernel that can be computed for non exhaustive interleaving rankings, the Kendall kernel, is not able to scale to larger dataset, as shown by the empirical time complexity analysis in Figure 6(d). Also note that the claim of the sushi dataset being state-of-the-art for evaluation is to be taken in the specific context of kernels methods for ranked data.
>
> ### Minor Clarifications
>
> > Indices in eq 18 seem to be incorrect.
> We have rechecked the indices in equation 18. I could not understand the issue with the indices. It would be helpful if you could please point to it again.
>
> Thank you for pointing out the mistakes. We have taken all minor suggestions into account in the revision of the paper.

---

> > ### Comment · Reviewer_moz1 · 2022-06-06
> > **Response**
> >
> > Thank you for your response.
> >
> > > The strictly increasing nature of the feature map is mainly due to the use of submodular functions.
> >
> > I understand the practical motivation. And it is indeed interesting that the presented empirical results are positive.
> >
> > I do not agree a ranking method that performs reasonably on a 5k sample dataset should be considered viable based on that sole result, especially if it scales badly as the authors have also pointed out in the draft. Having said that, there may be applications for which this could be viable. The TMLR guidelines only ask for a proof-of-concept. For that reason, I am leaning  towards acceptance.

---

### Review · Reviewer_iCnv · 2022-04-26

**Summary Of Contributions:**

The paper introduces a new feature representation of rankings, which is defined for both exhaustive and non-exhaustive partial rankings.
This representation is parameterized by a submodular function $F$ on a set of items $\cal V$. One of the main property of this representation is its sparing computation cost: ${\cal O}(pl)$ for an exhaustive ranking, where $p$ is the computation cost of $F$, and $l$ is the number of equivalence classes of the partial ranking.

Some experiments demonstrate the interest of the proposed representation for a supervised binary-classification task where examples are rankings on a finite set of objects $\cal V$.
* First, a similarity graph $G$ on items is built from item features and the submodular function $F$ is defined as the graph-cut function on $G$.
* Secondly, each example is associated to the feature representation derived from $F$.
* Finally, the classification task is handled by a Gaussian Process with linear kernel on-top of this feature representation.



**Requested Changes:**

The only critical adjustment is the correction of the errors in the text, figures and algorithms.

Regarding the experiments,

* I would prefer the experiments to be redone with a correct and more fair implementation,
* I would appreciate the addition of the naive baseline "the weight  of the $k$-th item is $n-k$",
* and I would appreciate the use of more standard Machine Learning algorithm.

But I guess it would not change the conclusions of the experiments, so do not consider this changes as mandatory.

I would also prefer the positioning of the paper to be updated, but it's also not mandatory. Note that if the positioning of the paper is changed, the title of the paper has to change too.
On the other hand, if the positioning remains the current one, please give a complete definition somewhere (Maybe in Section 3.3).


**Strengths And Weaknesses:**

# Significance and impact

> The paper proposes a new feature representation for rankings with many potential applications. However the theoretical properties of this representation are not discussed and the experimental section is not exploring the impact of these properties. The application of the proposed representation may prove to be limited to rare cases.

My main concern regards the fact that a set of extremely different rankings may share the same representation. Indeed, as soon as the set of constraints $s(B_i)\leqslant F(B_i)$ is made of large thresholds $F(B_i)$, the point $\bar s := \frac{1}{|{\cal V}|}[1,\dots,1]$ belongs to the Tangent cone and $\Phi(A)=\bar s$. I wonder to which extend this is a bad behavior with respect to the targeted machine learning task.

It seems that with graph-cut submodular function, these rankings are the one which are incoherent with the inherent clustering of items. Such rankings should be unfrequent in the dataset, and anyway they cannot be classified by any model.

Similarly, the paper would be more convincing if it would give some theoretical insights on the link between the graph $G$ and the corresponding feature space and/or linear kernel.

# Clarity
> While the paper is well-written and clear, it suffers several mistakes.

First, Kumar & Bach (2017) where considering ordered partitions presented in decreasing order, while current paper use the opposite convention. Unfortunately, some Equations and Algorithm 1 are not coherent with this choice. Here are the errors I have spotted, but I may have missed some others:

* in Section 2.4 and in algorithm 1, $B_i$ should be the union from $i$ to $l$
* in Equation (9), the constraint should be $v_1 \leqslant \dots \leqslant v_l$.

Similarly, the labels of Figure 1a are incoherent with the labels of Figure 1b.

Secondly, current version of Algorithm 1 is of complexity ${\cal O}(l^2)$ due to the "computation from scratch" of $B_{k-1}$ and $B_k$.

Finally, the clarity could be improved with some minor changes or additions:

* In Section 2.4, remind that $sup_{s\in \hat B^A(F)} ws^T = f(w)$ or $\infty$ depending on $w$.
* Close to Equation (9), remind that $B_0 = \emptyset$.
* In section 3, define $m$.
* In algorithm 1, "Define a*** set to contain ..."
* From a figure to another, the proposed approach is denoted *Our*, *Ours* or *Cut (Ours)*. Please be coherent.
* In figure 6b, the y-axis is increasing, while in Figures 6c and 6d it is decreasing. Please be coherent.
* The y-axis of Figure 6c is missing its label.
* The experimental setting would be easier to understand if the classification model would be more usual (SVM, random forest, logistic regression, ...).
* In Section 5, the second sentence of the second paragraph is missing a verb.


# Accurate evaluation
> The interest of the proposed representation is only supported by experiments.

> These experiments and the corresponding results are interesting, but the experiments remain frugal and there is some incoherences in the shared code.

First, while the proposed representation may be used in many ways, the experiments limit themselves to one submodular function, one kernel, and one classification model.

Secondly, the representation designed by the proposed approach consists in weighting items, with an higher weight for top-ranked items. We may design a trivial representation with the same property: associate the $k$-th item to the weight $n-k$. What would be the results obtained with such naive representation?

Thirdly, in current implementation, Mallows and Kendall-Tau kernels are obtained by computing the appropriate kernel on-top of a representation of the examples in a $n(n-1)/2$ feature-space. I wonder to which extend such indirect computation may artificially increase the computation time of corresponding experiments.

Finally, current implementation of the proposed representation

* differs from an experiment to another,
* is wrong for full_and_topk experiments (constraint violations are not handled),
* is quadratic
    * due to the `B_i_ = B_i.copy()`, L.56 in full_and_topk
    * due to `reduce(lambda a, b: a.union(b), ...)`, L.113 and L.114 in interleaving


# Positioning (clarity bis)
> In my opinion, current positioning of the paper is not the right-one, and hinders the clarity of the paper.

Let me first remind the official positioning (given paper's title, abstract and introduction): "A new and efficient to compute kernel for ranking based on the graph cut function". In my opinion, this positioning is misleading in two aspects:

* The linear kernel is not required to use the proposed representation.  Any standard Machine Learning model may be used on-top of it.
* A graph cut function is not required to define the proposed kernel, any submodular function would do the job. However, I appreciate that the paper demonstrates the utility of this kernel through a graph cut function, as such functions are easy to define in many applications.

The discrepancy between what brings current paper and what is claimed at the beginning of the paper hinders its clarity. The technical sections (2 and 3) are defining a family of kernels, but not the graph-cut one:

* given a submodular function $F$, Sections 2 and 3.1 explain how to get a representation;
* given a representation, Section 3.2 explains how to get a kernel;
* given a submodular function $F$ of complexity ${\cal O}(p)$, Section 3.3 gives the overall complexity of the proposed approach (it also reminds the PAVA algorithm, which is the one used to get the representation).

This family of kernel is used in combination with the graph cut function only in experiments (and in figures 2 and 3).

To resume, the pitch mentions a "graph-cut kernel", but the technical part presents a much more general framework, and this framework is used to get a "graph-cut kernel" only in applications. It's up to the reader to guess that it will be so.

And the worst is that it has to be a guess. The fact that "graph-cut kernel" means "the proposed kernel applied with a $F =$ graph-cut function" has to be inferred from the reader, it is neither mentioned. the closest sentence are
* "In Section 2, we show how to encode all kinds of rankings as ordered partitions and their relationship with submodular functions, in particular with the graph cut function" (Section 1)
* "Optimal values $v^*$, Eq. (11), using the submodular cut function and the information graph of the sushi dataset for $A$." (legend of figure 2)
* "In our approach, we use graph cut functions that shows a principled way to represent the permutohedron through its base polytope." (Section 5 Discussion)

---

> ### Author Response · Authors · 2022-05-16
> **Significance and impact**
>
> > My main concern regards the fact that a set of extremely different rankings may share the same representation.
>
> This is a valid concern when the submodular functions are trivial, such as concave functions on cardinality of the set or a modular function. The example given by the reviewer fits this setting. The tangent cone of base polytope for such functions does not drastically change the min-norm point. However, in our setting, we specifically are interested in graph cut functions that are not trivial and built using the numerical variables that represent the underlying objects (in our case different sushis). Therefore, the min-norm points are very different for all the experiments that we have shown. This representation *may* be the same if the corresponding ordered partitions are close for instance $1 < 2 < 3 < 4$ and $1 < \{2,3\} < 4$. However, if the tangent cones are $1 < 2 < 3 < 4$ and $4 < 3 < 2 < 1$, then the tangents cones are very different. It is this characteristic that is useful for defining kernels.

---

> ### Author Response · Authors · 2022-05-16
> **Clarity**
>
> > Feature Representation.
>
> Kindly note that it is only a change of representation from $l$ to $k$ to define $B_k$ instead of $B_l$. Please note that the constraints are correct. When we get the feature representation, which is a dual optimal of the primal solution, the inequality constraints change direction. The optimal $v$ is strictly decreasing, so that the optimal $s$ (feature map) is strictly increasing. In our algorithm, we only make $l$ function evaluations of $F(B_j)$ for all $j = 1 \ldots l$. Kindly consider this as a look up table. The PAVA algorithm thus has only $O(l)$ runtime complexity.
>
> > Positioning of the paper.
>
> We have favour the graph cut function because, contrary to other submodular function such as the entropy or the mutual information, it can be computed efficiently. We have introduced section 2.4 on Graph cuts and why we favour graph cuts over general submodular functions.
>
> Thank you for pointing out the mistakes. We will improve the clarity of the paper. We have taken all minor suggestions into account in the revision of the paper.

---

> > ### Comment · Reviewer_iCnv · 2022-05-16
> > **runtime complexity**
> >
> > I'm pointing toward the quadratic runtime complexity due to the handling of $B_j$ (I'm aware that the computation of each $F(B_j)$ is put apart in the $O(lp)$ term). For each of your proposals [1, 2, 3], this handling has a quadratic complexity (while it exists implementations with a linear complexity).
> >
> > I would not argue for rejecting the paper on this basis, especially as for most submodular functions $p=\Omega(n)$, but such mistakes are frustrating for someone with a computer science background.
> >
> > [1] definition of PAVA in the original version of the paper
> > $B_k \to \bigcup_{i=1}^kA_i$ costs $|\bigcup_{i=1}^kA_i|=O(n)$ if implemented basically, which leads to a $O(ln)$ computation cost for the full loop.
> >
> > [2] your implementation of PAVA
> > `reduce(lambda a, b: a.union(b), ...)` costs $|\bigcup_{i=1}^kA_i|=O(n)$, which leads to a $O(ln)$ computation cost for the full loop.
> >
> > [2] definition of PAVA in May, 16 version of the paper
> > "Initialize $B_1,\dots , B_l$ // using Eq. (7)" costs $O(ln)$ if implemented and stored basically.

---

> > > ### Author Response · Authors · 2022-05-17
> > > **Updated Complexity**
> > >
> > > Thank you for clarifying and pointing out our mistake. We agree that it is $O(ln)$ complexity and we have updated the complexities of computing the Gram matrix accordingly.

---

> ### Author Response · Authors · 2022-05-16
> **Accurate Evaluation**
>
> > In current implementation, Mallows and Kendall-Tau kernels are obtained by computing the appropriate kernel on-top of a representation of the examples in a $n(n-1)/2$ feature-space. I wonder to which extend such indirect computation may artificially increase the computation time of corresponding experiments.
>
> The computing timing reported for the Mallows and Kendall kernels are not calculated using the implementations in the shared code but using the code from https://github.com/YunlongJiao/kernrank, which contains the official implementation of the Kendall kernel and an efficient implementation of the Mallows kernel. The reason we use the $n(n-1)/2 feature space is to leverage GPFlow and thus to speed up computations using the GPU for the experiments. Note that this is only possible for experiments with low number of items, so not for the experimented in Figure 5(d).
>
> > is wrong for full_and_topk experiments (constraint violations are not handled),
>
> The code from L.48 to L.58 is part of an initial phase of the project when we were testing the impact of constraints violations. We will update the shared code to remove those lines of code and so to reflect that the code used for the calculation is L.73 to L.85, which the constraints violations are handled and is identical to the code in the interleaving experiments.
>
> > is quadratic.
>
> Although better non-quadratic implementation may be possible, the practical cost of using reduce(lambda a, b: a.union(b), ...) is negligible.
>
> > and I would appreciate the use of more standard Machine Learning algorithm.
>
> We favoured Gaussian Processes over SVMs since they have already been employed in the literature of kernels for ranked data. See Lomeli, Maria, et al. "Antithetic and Monte Carlo kernel estimators for partial rankings." arXiv preprint arXiv:1807.00400 (2018)..

---

### Review · Reviewer_w9xB · 2022-05-02

**Summary Of Contributions:**

The paper deals with kernels for ranking data. The authors introduce the known connection between submodular functions (here the graph cut function) and the base polytope (Lovász, 1982, Rockafellar, 1997).  The connection to ordered partitions, and thus rankings, arises since the faces of the base polytope are characterized by ordered partitions. Since the number inequalities to describe the base polytope scale exponentially with the ground set (to be ranked), the authors resort to a standard outer approximation based on the tangent cone, requiring a linear number of inequalities.

Based on the above approximation they define the feature map of the graph cut kernel as the min-norm point of the tangent cone, which can be computed via isotonic regression (Kumar & Bach, 2017). The isotonic regression problem can be computed using the PAVA algorithm (Best & Chakravarti, 1990), whose solution induces the needed ranking.

Further, the authors, show how the kernel can be easily extended to non-exhaustive partial orderings via Haussler's convolutional kernel, see Eq. 13, and present a simple sampling scheme to speed up the computation of the kernels.

Finally, the authors report empirical evidence on two datasets (one of them synthetic) for various ranking settings, using a Gaussian process classifier, showing that their method is slightly better than the two baseline approaches while reporting lower computation times.





**Broader Impact Concerns:**

I do not see any ethical implications of the work that would require adding a Broader Impact Statement.

**Requested Changes:**



**Major points**

Given the incremental nature of the paper, the authors should invest more work into the experimental section, i.e.,
* conduct experiments on more datasets, especially large-scale ones, showing a clear computational benefit over the baselines
* Conduct experiments in all three settings of Table 1
* The (submodular) graph cut function seems to be never introduced formally
* Make more clear why you focus on the graph cut function

**Minor points**
* Page 3, also explain exhaustive ranking, the ordered partition in Table 2 might not be clear immediately
* Page 3, Eq. 2., add some intuition for those not familiar with submodular functions.
* Algorithm 1 is hard to parse, think of a better presentation

**Questions:**
- Can anything be said about the quality of the rankings induced by the original base polytope versus its outer approximation?
- How did you pick the number of samples for the method described in Section 3.2 in the experiments?
- What are the challenges when applying the method to other submodular functions beyond the cut function.
- Is there a reason to use Gaussian processes over other classifiers, e.g., SVM.

**Strengths And Weaknesses:**

* Strengths
  * Methods work for different kinds of ranking data, cf. Table 1.
  * Well-written, most parts are easy to follow
  * Review of related work seems adequate

* Weakness
  * The methodological contributions are rather weak or *incremental*, that is, the kernel computation is based on known, established techniques, which are only applied here.
  * Given the above incremental contribution, the experiential evaluation is not very extensive and restricted to two (small) datasets, one synthetic and small (ground set has cardinality **8**).

---

> ### Author Response · Authors · 2022-05-16
> **Additional experiments**
>
> > Conduct experiments in all three settings of Table 1.
>
> Would the reviewer please clarify which setting of Table 1 has not been addressed? Figures 4(a) and 5(a) correspond to the full setting, Figures 4(b), 4(c) and 5(b) correspond to the top-k setting, Figures 4(d) and 5(c) corresponds to the exhaustive interleaving setting and Figure 5(d) correspond to the non-exhaustive interleaving setting.
>
> > conduct experiments on more datasets, especially large-scale ones, showing a clear computational benefit over the baselines.
>
> The large version of the sushi dataset (5000 samples and 100 items) is the large scale dataset we have used (see Fig. 5(d)). We did not employ other large scale datasets because the only state-of-the-art kernel that can be computed for non exhaustive interleaving rankings, the Kendall kernel, is not able to scale to larger dataset, as shown by the empirical time complexity analysis in Figure 6(d).

---

> ### Author Response · Authors · 2022-05-16
> **Contribution**
>
> > The methodological contributions are rather weak or incremental, that is, the kernel computation is based on known, established techniques, which are only applied here.
>
> The state of the art kernels do not use any form of structural information from the dataset. The reason we improve computationally considerably is because we have been able to use additional information that can be encapsulated in the form of graphs, which in turn helps us design efficient kernels. This approach has not been attempted earlier with kernels for ranked data.
>
> > Make more clear why you focus on the graph cut function.
>
> We have favour the graph cut function because, contrary to other submodular function such as the entropy or the mutual information, it can be computed efficiently. We have introduced section 2.4 on Graph cuts and why we favour graph cuts over general submodular functions.
>
> > Can anything be said about the quality of the rankings induced by the original base polytope versus its outer approximation?
>
> Every ordered partition (uniquely represented by a ranking) gives an outer approximation of the base polytope. The min-norm point on the base polytope is unique in the absence of ranking. We have not been able to show any theoretical distance of how the min-norm point moves based on the ordered partition.
>
> > How did you pick the number of samples for the method described in Section 3.2 in the experiments?
>
> The number has been manually tuned to balance the computational cost of the procedure and the f1-score obtained on a train set.
>
> > What are the challenges when applying the method to other submodular functions beyond the cut function?
>
> The main challenge is that, in general, submodular functions are expensive to compute when compared to graph cuts.
>
> > Is there a reason to use Gaussian processes over other classifiers, e.g., SVM?
>
> We favoured Gaussian Processes over SVMs since they have already been employed in the literature of kernels for ranked data.
>
> > Algorithm 1 is hard to parse, think of a better presentation.
>
> We improved the presentation of the PAVA algorithm.

---

### Decision · Action_Editors · 2022-06-18

**Recommendation:** Accept as is

**Comment:**

The reviewers judged this paper as presenting a potentially promising approach with a proof of concept empirical section. While the empirical section was deemed a bit weak, the reviewers judged that it was sufficient to illustrate the potential of the approach, which could be investigated more thoroughly by the community. All reviewers recommended acceptance and thought it was of interest to the TMLR community.

As the authors have already addressed most of the corrections asked by the reviewers in their revision, the paper can be accepted as is. Any missing clarification can be included in the camera ready version (please detail in a comment on Openreview any other changes made to the final version of the paper and how it relates to the reviews).